

# Autonomous Sensory Meridian Response self-reporters showed higher scores for cognitive reappraisal as an emotion regulation strategy

Ricardo Morales[1,2], Daniela Ramírez-Benavides[1] and
Mario Villena-Gonzalez[1,3]

[1] Escuela de Psicología, Pontificia Universidad Católica de Chile, Santiago, RM, Chile
[2] Center for Cognitive Neuroscience, Duke University, Durham, NC, United States of America
[3] Centro de Estudios en Neurociencia Humana y Neuropsicología, Facultad de Psicología,
Universidad Diego Portales, Santiago, RM, Chile

Corresponding author
Mario Villena-Gonzalez,
mvillena@ug.uchile.cl

## ABSTRACT

**Background:** Autonomous Sensory Meridian Response (ASMR) describes the experience of a pleasant tingling sensation along the back of the head, accompanied with a feeling of well-being and relaxation, in response to specific audio-visual stimuli, such as whispers, soft sounds, and personal attention. Previous works have assessed individual variations in personality traits associated with ASMR, but no research to date has explored differences in emotion regulation associated with ASMR. This omission occurred even when ASMR, a sensory-emotional experience, has been proposed to be located in a sound sensitivity spectrum as the opposite end of misophonia, a phenomenon associated with difficulties regulating emotions. The present work aimed to assess group differences between ASMR self-reporters and non-ASMR controls associated with emotion regulation strategies.
**Methods:** We used the validated Spanish version of the Emotion Regulation Questionnaire to assess individual differences in the use of cognitive reappraisal and expressive suppression.
**Results:** Our results showed that participants who experience ASMR had higher scores in the cognitive reappraisal subscale of the emotion regulation questionnaire than the non-ASMR group.
**Conclusions:** Individuals who experience ASMR reported higher use of cognitive reevaluation of emotionally arousing situations, suggesting more effectiveness in regulating emotions. Our finding further elucidates individual differences related to this experience, supporting that ASMR is a real psychophysiological phenomenon associated with other psychological constructs and has remarkable consequences in affective/emotional dimensions and general well-being.

## INTRODUCTION

Autonomous Sensory Meridian Response (ASMR) describes the experience of a pleasant tingling sensation along the back of the head and neck region, at times spreading to other areas of the body, in response to specific visual and auditory stimuli (*Barratt & Davis, 2015*). Usually, ASMR is triggered by whispers, soft sounds made with the fingers on a surface, and personal attention from someone with an affectionate disposition (*Barratt, Spence & Davis, 2017*; *Fredborg, Clark & Smith, 2017*; *Poerio et al., 2018*).

This non-scientific term "ASMR" was coined just in 2010 (*del Campo & Kehle, 2016*; *Morris, 2018*). During the last decade, social networks have made it possible to publicize this phenomenon and quickly disseminate different personal reports on this experience, along with the creation of videos ("ASMR videos") that simulate or accentuate the stimuli to trigger the sensation in people who watch/listen to them (*del Campo & Kehle, 2016*). Anecdotal accounts and early research have shown that ASMR seems to be experienced by only part of the population, and therefore some individuals do not experience it (*Barratt & Davis, 2015*; *del Campo & Kehle, 2016*). There are no studies to date about the prevalence of ASMR-capability in the general population; however, some studies have offered some estimates suggesting the phenomenon is widespread (*Poerio, 2016*; *Roberts, Beath & Boag, 2020*). ASMR's popularity on the internet supports how prevalent this phenomenon might be (Fig. S1).

Scientific investigation about this experience started even more recently, with the first peer-reviewed paper on the topic published in 2015 (*Barratt & Davis, 2015*). Since then, there has been a growing interest in deciphering this phenomenon's underlying psychological and biological basis (Fig. S2).

Some previous works have assessed individual variations in personality traits associated with ASMR. These works have shown that, compared with non-ASMR controls, ASMR self-reporters scored higher on openness to experience (*Fredborg, Clark & Smith, 2017*; *McErlean & Banissy, 2017*), which is associated with curiosity about the world around them and propensity to have vivid fantasies or daydreams (*John, Naumann & Soto, 2008*). In line with these results, other works have shown ASMR self-reporters score higher on Fantasizing (*McErlean & Banissy, 2017*), reflecting an ability to immerse oneself in a fictional reality (*Davis, 1983*). They also produced higher scores on the Curiosity subscale of the Toronto Mindfulness Scale (TMS), suggesting a greater interest in and openness to their own conscious experiences (*Fredborg, Clark & Smith, 2018*). Individuals with ASMR also showed higher scores on neuroticism, associated with increased self-consciousness (*Fredborg, Clark & Smith, 2017*). Finally, ASMR self-reporters have shown to score higher on absorption (*McErlean & Osborne-Ford, 2020*; *Roberts, Beath & Boag, 2019*), which is the ability to get deeply immersed with the current experience accompanied by loss of reflective awareness, and it has been previously linked to imagery, daydreaming, and openness to experience (*Glisky et al., 1991*). All these results together suggest that an increased tendency to have openness to conscious experience, fantasize, and imaginatively transposing oneself into a virtual reality may be a key feature of ASMR responders.

Despite the notorious and growing interest in the subject, no research has explored the individual variations of emotional-related traits associated with ASMR, excepting some works assessing the relation between empathy and ASMR (*McErlean & Banissy, 2017*; *Poerio et al., 2018*). The lack of studies directly evaluating emotional-related traits has occurred even when the literature is continuously emphasizing that ASMR is a sensory-emotional experience rather than just a sensory one (*Fredborg, Clark & Smith, 2018*; *Smith, Fredborg & Kornelsen, 2019*; *Smith, Katherine Fredborg & Kornelsen, 2017*).

Previous works have revealed the emotional dimension of ASMR by evaluating self-report measures, neuroimaging, and physiology. For instance, according to the first study on the topic, 80% of the participants reported watching ASMR videos because it positively affects their mood, usually accompanied by a pleasant feeling of well-being and relaxation (*Barratt & Davis, 2015*). With regard to the neuroimaging approach, activation in regions related to emotional arousal (dorsal anterior cingulate cortex, insula, and inferior frontal gyrus) has been shown using fMRI during tingling sensations (*Lochte et al., 2018*). Another study measured physiological responses comparing ASMR self-reporters and non-ASMR controls while watching ASMR videos. The results showed a significant reduction in heart rate and increased skin conductance levels in ASMR self-reporters compared with non-ASMR controls suggesting that ASMR is a complex emotional experience blending different emotional components such as relaxation and arousal (*Poerio et al., 2018*).

In line with these previous studies supporting ASMR as an emotional experience, ASMR has also been suggested to be related to another underreported condition known as misophonia, where familiar repetitive sounds, mostly sound produced by humans such as chewing and sniffing, may trigger negative physiological, emotional, and behavioral responses (*Roberts, Beath & Boag, 2019*; *Rouw & Erfanian, 2018*).

Previous works have suggested that misophonia and ASMR might represent two ends of the same spectrum of sound sensitivity where human-generated sounds elicit pleasurable tingling sensation in case of ASMR and negative physical and emotional responses in case of misophonia (*Barratt & Davis, 2015*; *McErlean & Banissy, 2018*).

Previous works have shown that individual variations in the severity of misophonia symptoms positively correlate with emotional regulation difficulties (*Cassiello-Robbins et al., 2020*). Neuroimaging approaches to misophonia revealed abnormal functional connectivity between the anterior insular cortex (a hub of the salience network critical for interoception and emotion processing) and brain regions responsible for the regulation of emotions (*Kumar et al., 2017*).

It is interesting to note that even when misophonia and ASMR have been proposed as opposite poles on the same spectrum, they also have been reported to overlap and co-occur in the same individuals, but the experience may depend on the trigger stimuli or context (*Roberts, Beath & Boag, 2019*; *Rouw & Erfanian, 2018*).

ASMR has also been widely associated with another related sensory-emotional phenomenon known as frisson or music chills (*del Campo & Kehle, 2016*). Frisson is a chill down the spine sensation that occurs while engaged in enjoyable music and other emotional experiences and is generally described as a pleasant sensation with an emotional

load; Tears, gooseflesh, or piloerection could accompany it (*Harrison & Loui, 2014*). Neuroimaging studies have shown that brain activation during ASMR showed similarities to patterns previously observed in music chills, associated with the striatal reward pathway and emotional arousal regions (*Lochte et al., 2018*; *Salimpoor et al., 2011*). These similarities can also be found in electrophysiological measures, in which both phenomena elicit an increase in skin conductance (*Poerio et al., 2018*; *Salimpoor et al., 2011*).

Nonetheless, essential differences have been observed between these phenomena involving activation of the medial prefrontal cortex (mPFC). This region showed increased activation during ASMR tingling sensation, which has not been observed for music chills (*Lochte et al., 2018*). Activation of mPFC has been associated with higher scores in empathy and emotional regulation capabilities (*Esménio et al., 2019*; *Tang, Tang & Posner, 2016*), especially for reappraisal (*Nelson et al., 2015*). In line with this, previous research has shown that ASMR self-reporters have higher Empathic Concern levels (*McErlean & Banissy, 2017*). This trait has been previously associated with higher scores in cognitive reappraisal (*Laghi et al., 2018*; *Lockwood, Seara-Cardoso & Viding, 2014*).

Taking this evidence into account, it would be logical to assume that the ability to feel ASMR could be associated with differences in some facet of emotional regulation, for instance, the regulation strategy deployed (suppression or reappraisal). However, there is no evidence about ASMR/non-ASMR group differences related to the emotional regulation abilities to date.

Individual differences in emotion regulation mechanisms can be investigated by studying the most common strategies: suppression and reappraisal (*Gross & John, 2003*). Suppression is a form of response modulation in which the ongoing emotion-expressive behavior becomes inhibited. Reappraisal involves a cognitive reevaluation of the emotionally arousing situation to alter its emotional impact. Research on this topic has shown that using reappraisal is more effective and related positively to well-being, whereas using suppression is negatively related (*Gross & John, 2003*; *Morawetz, Alexandrowicz & Heekeren, 2017*). Previous studies demonstrated an association between reappraisal use and personality traits such as neuroticism and openness to experience (*Morawetz, Alexandrowicz & Heekeren, 2017*; *Wang, Shi & Li, 2009*). These same traits have also been strongly associated with people who experience ASMR (*Fredborg, Clark & Smith, 2017*; *McErlean & Banissy, 2017*).

The present work aims to assess group differences between ASMR self-reporters and non-ASMR groups associated with emotional regulation strategies. We hypothesized that people who experience ASMR, in the same way they seek to listen/watch ASMR stimuli to improve their positive affect and well-being, use an emotional regulation strategy that better provides effectiveness regarding well-being in different situations. We predicted that being the opposite extreme of misophonia (which is a condition related to difficulties in emotional regulation), people who experience ASMR will show greater cognitive reappraisal use than non-ASMR self-reported controls. This strategy has been positively associated with well-being and used by people who share personality traits common to ASMR groups. Finally, this prediction is also supported by the ASMR self-reporters'

known capabilities regarding fantasy/absorption, which could facilitate the propensity to change the way they think about emotional situations.

We used the validated Spanish version of the Emotion Regulation Questionnaire (*Cabello et al., 2013*) to assess individual differences in the use of cognitive reappraisal and expressive suppression in ASMR self-reporters and non-ASMR controls.

This is the first study to examine whether ASMR self-reporters show differences associated with emotional regulation strategies compared with non-ASMR controls. Therefore, this work further elucidates individual differences related to this phenomenon and its remarkable consequences in affective/emotional dimensions and well-being.

## MATERIALS & METHODS

### Participants

The total number of participants who answered the online survey was 177. From this total, only 69 reported not experiencing ASMR. One of these 69 participants answered the same response to all of the questions in the cognitive reappraisal and expressive suppression subscales. Due to this pattern of responses, we decided to exclude this participant's data from further analysis. Our sample consisted of 108 participants who report to experience ASMR and 68 participants who do not report to experience ASMR.
All participants were native Spanish speakers. The matched samples' nationalities were mostly Chilean; 167 Chilean, two Argentinian, three Spanish, two Venezuelan, one Mexican, and one Peruvian.

Due to the high proportion of participants who reported experiencing ASMR, we took a random subsample of 68 participants from 108 who declared experiencing ASMR to attain similar sample sizes. The matched samples consisted of 136 volunteers; 68 participants in the ASMR group (51 female, mean age = 22.40, SD = 3.32) and 68 participants in the non-ASMR group (52 female, mean age = 23.81, SD = 5.51). Groups did not differ in age ($t(134) = 1.81$, $p = 0.07$) nor in the proportion of male to female participants ($\chi^2 (1) < 0.01$, $p = 1$). Analysis with the total sample of the participants can be found in the Supplemental Materials. The participants were not rewarded for participating in this study.

### Procedure

Participants were recruited by an online invitation to participate in the present work. This invitation was spread through social media, specifically through Instagram and Facebook, posted in ASMR groups and also in groups non-related to ASMR (e.g., University students groups on Facebook). The invitation explained the study without giving away information related to ASMR to avoid bias towards people who might know and were interested exclusively in ASMR.

Participants were asked to complete a survey hosted in Google forms. This online survey included questions about personal information (age and gender) followed by the Spanish version of the Emotion Regulation Questionnaire (*Cabello et al., 2013*). Then, they

were provided with the Spanish version of the ASMR-15 Score questionnaire (*Roberts, Beath & Boag, 2019*).

Afterward, participants were asked if they knew what Autonomous Sensory Meridian Response is. Regardless of the answer, they then were provided with a description and question about experiencing ASMR. Specifically, participants were provided with the following statement: "ASMR is defined as a pleasant sensation of 'tingles' across the back of the head and neck in response to specific visual and auditory stimuli." Then, participants were asked to answer the following question, "would you defined yourself as someone who experiences ASMR?" with a Yes or No response. We divided the ASMR self-reporters and non-ASMR groups based on this response. Additionally, we further asked the ASMR group to provide their favorite ASMR video link/channel and answer some additional questions about the frequency of watching ASMR videos/content and the main reason for watching them if that was the case (see Supplemental Material).

The protocol was approved by the Ethics Committee of Pontificia Universidad Católica de Chile (approval reference number: 190325015). All participants gave electronic informed consent following the Declaration of Helsinki. This study was conducted online, and participants completed the questionnaires in their own time in one sitting.

## Materials

### Emotion Regulation Questionnaire (ERQ)

The ERQ is a self-report questionnaire that consists of 10-items assessing two different regulation strategies. The first one is called cognitive reappraisal based on six items, and the second one is called expressive suppression based on four items. Cognitive reappraisal focuses on the person's attempts to change the emotional impact by changing how they think of the situation. Expressive suppression focuses on the person's attempts to hide or inhibit the expression of their emotions. The instruction was: "We would like to ask you some questions about your emotional life, in particular, how you control (that is, regulate and manage) your emotions. We are interested in two main aspects: on the one hand, your emotional experience, or how you feel the emotions; and on the other hand, the emotional expression or how you show emotions in the way you talk or behave. Although some of the following questions may seem similar, they differ in quite important ways. Please answer how much you agree or disagree with each of the questions below using the following response scale". Both subscales are measured with simple questions answered on a Likert scale ranging from 1 (totally disagree) to 7 (totally agree). The higher scores signify more usage of the strategy they are referring to in that specific question. In this case, the questionnaire used was the Spanish adaptation of the ERQ, which has been translated and validated for Spanish-speaking users, showing good internal consistency for both subscales; expressive suppression (Cronbach's alpha = 0.75) and cognitive reappraisal (Cronbach's alpha = 0.79) (*Cabello et al., 2013*). This questionnaire's internal consistency calculated from data of the present study is similar to those reported in *Cabello et al. (2013)*, with a Cronbach's alpha = 0.76 for cognitive reappraisal and Cronbach's alpha = 0.78 for expressive suppression.

### ASMR-15 SCORE

The ASMR-15 is a multidimensional self-report measure of ASMR propensity that consists of 15-items assessing four different characteristics of the phenomenon; sensation, relaxation, affect, and altered consciousness (*Roberts, Beath & Boag, 2019*). "Sensation" describes aspects of the location and physical sensation. "Relaxation" involves changes in arousal associated with calm and relaxation. "Affect" describes aspects of emotional experience. Finally, "Altered Consciousness" is associated with shifts in perception and awareness. In this questionnaire, five items are related to sensation, three related to relaxation, four related to altered consciousness state as a result of the ASMR experience, and three others are related to affect, specifically to changes in emotional states. Participants answer with a Likert scale ranging from 1 (totally disagree) to 5 (totally agree), where higher scores indicate a greater tendency to experience ASMR. We used this scale as a global corroboration that the selection of groups based on self-report was different on the different dimensions of the phenomenon.

The scale has a brief introduction: "This survey is looking at how certain stimuli affect you. Some individuals experience intense physical and emotional responses upon hearing particular sounds. These sensations and feelings can be pleasant or unpleasant. Sounds such as whispering, crackling, tapping, or scratching may produce particular experiences described below. Using the scale, please indicate your level of agreement with each statement, upon hearing any of these, or similar sounds. When I hear certain sounds, such as whispering, crinkling, tapping…".

For this questionnaire, our research group translated the original instrument to a Spanish version, which showed good levels of internal consistency in the four subscales. The sensation subscale had a Cronbach's alpha = 0.81, the relaxation subscale had a Cronbach's alpha = 0.88, the affect subscale had a Cronbach's alpha = 0.80, and the altered consciousness subscale had a Cronbach's alpha = 0.83. The total score (the mean of the 15 items) had a Cronbach's alpha = 0.90.

## Statistics

All the data analysis, processing, basic descriptive statistics, testing of assumptions, and comparison between means were performed using the R and RStudio software (*Allaire, 2012*). Reliability analysis of the subscales of the ERQ and ASMR-15 was made with the psych package (*Revelle, 2015*) and plots with the ggplot2 package (*Wickham, 2011*).

To match the sample sizes between the ASMR and non-ASMR groups we randomly selected a subsample from the group with more participants, equal to the number of participants of the group with fewer participants. We repeated this procedure 1,000 times to corroborate the replicability of the effect. To test if the ASMR and non-ASMR groups differ in age or male to female participants, we performed a *t*-test and a chi-square test, respectively. Subsequently, we used the Cronbach alpha coefficient to measure the internal consistency of the subscales of the ERQ and the ASMR-15. All of the comparisons between the ASMR and non-ASMR groups were performed with independent sample *t*-tests.

**Table 1 ASMR and non-ASMR scores in the emotion regulation questionnaire.**

| Scale | Group | | | | | |
| | ASMR | | | Non ASMR | | |
| | Mean | Median | SD | Mean | Median | SD |
|---|---|---|---|---|---|---|
| Cognitive reappraisal | 4.69 | 4.92 | 1.14 | 4.19 | 4.17 | 1.16 |
| Expressive suppression | 3.53 | 3.50 | 1.28 | 3.73 | 3.5 | 1.51 |

## RESULTS

Our main aim was to assess whether ASMR self-reporters show differences in emotional regulation strategies compared to the non-ASMR group. We performed individual comparisons between each of the subscales of the questionnaire. To maintain the experiment-wise error rate to 5%, we set an alpha value of 2.5% (a significant $p$-value of 0.025).

### Normality and homogeneity of variance

To test the assumptions of normality and homogeneity of variance, we use the Shapiro–Wilk and Levene tests, respectively. In the Expressive suppression subscale, data from the ASMR group ($W = 0.98$, $p = 0.41$), and data from the non-ASMR ($W = 0.97$, $p = 0.07$) showed a normal distribution. In the Cognitive reappraisal subscale, both the ASMR group ($W = 0.97$, $p = 0.11$) and the non-ASMR group ($W = 0.98$, $p = 0.37$) showed a normal distribution. The variance between the ASMR and non-ASMR groups were similar in both the Expressive Suppression subscale ($F(1,134) = 2.31$, $p = 0.13$) and in the Cognitive Reappraisal subscale ($F(1,134) = 0.11$, $p = 0.76$). In neither group, participants with z scores bigger than 3 or lesser than −3 were detected; accordingly, we didn't reject any case (*Tabachnick & Fidell, 2013*).

As the data followed a normal distribution and had equal variances, we used an independent $t$-test to compare groups. We applied a one-tailed $t$-test for the cognitive reappraisal subscale, as we predicted higher scores for the ASMR group. We used a two-tailed $t$-test for the expressive suppression subscale because we had no directional prediction for group differences.

### ASMR group differences in the Emotion Regulation Questionnaire (ERQ)

The ASMR group showed significantly higher scores in the cognitive reappraisal subscale than the non-ASMR group ($t(134) = 2.53$, $p < 0.01$, Cohen's $d = 0.43$) (see Table 1). There were no significant differences between ASMR and non-ASMR groups for the expressive suppression subscale ($t(134) = −0.84$, $p = 0.40$). These results can be observed in Fig. 1.

We also performed the same analysis using all the sample and, after including the gender of the participants as a predictor, obtaining similar results (see Supplemental Material).

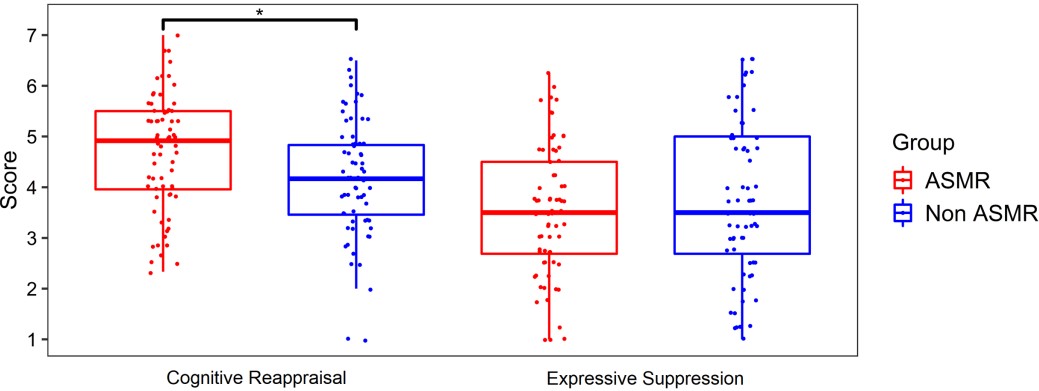

**Figure 1 Plot of ASMR and non-ASMR scores in the emotion regulation questionnaire.** Box-plot shows the scores of the two Emotional Regulation Questionnaire' subscales for the ASMR ($n = 68$) and non-ASMR ($n = 68$) groups. Group difference is significant only for the cognitive reappraisal subscale. Jittered points are individual subjects. An asterisk (*) indicates a significant difference ($p < 0.01$).

## Replicability of the effect using multiple random sub-samples

As our main analysis was performed on a subsample of 68 participants of the total sample who reported to experience ASMR (108 participants), we wanted to test the replicability and robustness of our effect by using multiple random sub-samples. Specifically, we repeated 1,000 times the procedure of randomly choose subsamples, each of 68 participants of the total of 108 participants who reported experiencing ASMR. We compared each subsample with the 68 participants who reported not experiencing ASMR. Of these 1,000 randomly chosen subsamples, we found that the ASMR group had significantly higher scores in the cognitive reappraisal subscale 84.60% of the time, binomial 95% CI [82.18–86.75%]. Regarding the expressive suppression subscale, we found no significant differences in 97.90% of the subsamples, binomial 95% CI [96.75–98.66%]. In a high percentage of subsamples, we found results similar to those from our main analysis, supporting the replicability of the effect in different subsamples of our data.

## Differences in ERQ scores using ASMR self-reporters who sought out ASMR videos

Given that group selection was based on self-report over the internet after a brief description of the phenomenon, naïve people may confound the experience (for instance, with frisson) and erroneously report themselves as ASMR experiencers. In order to have a more accurate selection of participants who experience ASMR we performed the same analysis as before, but only with the subjects who report to experience ASMR and also sought ASMR videos before. This new subsample was composed of 58 participants (45 female, mean age = 22.59, SD = 3.08). We compared this new subsample with the 68 participants who reported not experiencing ASMR. Groups did not differ in age ($t(124) = 1.50$, $p = 0.13$) nor in the proportion of male to female participants ($\chi^2 (1) = 0.09$, $p = 0.76$).

In the cognitive reappraisal subscale, the ASMR self-reporters who have watched ASMR videos had marginally significant higher scores than non-ASMR subjects ($t(124) = 1.66$, $p = 0.05$, Cohen's $d = 0.30$), with the ASMR group scoring 4.53 (SD = 1.12) and the non-ASMR group scoring 4.20 (SD = 1.16). In the expressive suppression subscale there were no significant differences between groups ($t(124) = -0.77$, $p = 0.44$).

## ASMR-15 score group differences

We used the ASMR-15 Scale as a global corroboration that the selection of groups based on self-report was different on several dimensions of the phenomenon. This analysis may help to confirm that participants who self-report to experience ASMR actually had felt it. Using the same sample from our main analysis, we wanted to test if participants who self-report to experience ASMR had higher scores in the four subscales of the ASMR-15, comparing with participants who do not report to experience ASMR.

Compared to the non ASMR group, the ASMR group had significantly higher scores in sensation ($t(134) = 6.03$, $p < 0.001$, Cohen's $d = 1.03$), relaxation ($t(134) = 5.20$, $p < 0.001$, Cohen's $d = 0.89$), affect ($t(134) = 5.08$, $p < 0.001$, Cohen's $d = 0.87$) and in altered consciousness ($t(134) = 3.59$, $p < 0.001$, Cohen's $d = 0.62$). Additionally, using the ASMR-15 total score (the mean of the 15 items), the ASMR group had significantly higher scores than the non ASMR group ($t(134) = 6.51$, $p < 0.001$, Cohen's $d = 1.12$). These results indicate that participants who self-report as ASMR experiencers, also scores higher in different dimensions involved in experiencing ASMR, providing further support to the group selection. Descriptive values of the subscales can be found in the Supplemental Material.

## DISCUSSION

The present study investigated group differences between ASMR self-reporters and non-ASMR groups associated with emotional regulation strategies. We predicted that people who experience ASMR would show more significant use of cognitive reappraisal compared with non-ASMR self-reported controls. Our results confirmed our prediction, showing with a moderate effect size that participants who experience ASMR had higher scores in the cognitive reappraisal subscale of the emotion regulation questionnaire than the non-ASMR group. This finding suggests that the ASMR group uses cognitive reevaluation of the emotionally arousing situation to alter its emotional impact to a greater extent than the non-ASMR group. On the other hand, both groups showed similar scores in the other subscale of the same questionnaire, suggesting no group differences for the use of inhibition of emotion-expressive behavior.

Our finding supports the notion that ASMR is closely associated with emotional behavior rather than merely a sensory response. As we mentioned before, a growing body of studies has explored how ASMR elicits emotional arousal, which can be observed at different levels, such as psychological/behavioral (*Barratt & Davis, 2015*), hemodynamical (*Lochte et al., 2018*), and electrophysiological (*Fredborg et al., 2021*; *Poerio et al., 2018*). We now show that this phenomenon is also associated with strategies that allow us to control (regulate and manage) emotions when facing emotionally arousing situations.

The higher use of the reappraisal strategy in people who experience ASMR is especially interesting given that reappraisal has been shown to be more effective to regulate emotions and related positively to well-being (*Gross & John, 2003*; *Morawetz, Alexandrowicz & Heekeren, 2017*). This contrasts with what has been shown for misophonia, suggested to be the opposite pole of ASMR, in which people have noticeable difficulties regulating emotions (*Cassiello-Robbins et al., 2020*).

We can speculate that the individual variations on reappraisal preferences might be associated with connectivity of brain networks associated with reappraisal, involving the orbitofrontal cortex selectively (*Kanske et al., 2017*). Studies in resting-state functional connectivity have reported that people who experience ASMR show the recruitment of the orbitofrontal cortex by sensory-motor networks, suggesting this is one neural substrate of ASMR's underlying emotional aspect. However, the differences reported in the present study may also be mediated by personality traits that have been independently related to ASMR and emotional regulation, such as neuroticism and openness to experience (*Fredborg, Clark & Smith, 2017*; *McErlean & Banissy, 2017*; *Morawetz, Alexandrowicz & Heekeren, 2017*; *Wang, Shi & Li, 2009*). As we did not measure these traits in our sample, future investigations may explore the specific relations between ASMR, personality traits, and emotional regulation.

While we showed that ASMR self-reporters obtained higher cognitive reappraisal scores, our study cannot shed light on the mechanisms responsible for this association. Follow-up studies should specify the relationship between experiencing ASMR and cognitive reappraisal strategies to regulate one's emotional states. We hypothesize that the higher tendency of ASMR self-reporters to deploy cognitive reappraisal strategies is mediated by their capacity to be able to immerse themselves in different types of experiences (*McErlean & Banissy, 2017*; *Roberts, Beath & Boag, 2019*), a capacity that might give them greater flexibility to change the way they think about and re-appraise emotional situations. If this hypothesis is correct, constructs related to the capacity to be absorbed by imagery, such as fantasizing capabilities and daydreaming (*Fox et al., 2013*; *Glisky et al., 1991*), should mediate the relationship between ASMR and cognitive reappraisal.

In the present study, we also found that almost half of the ASMR sample (46%) watch/listen to ASMR videos at least once a week (Fig. S4) which is in line with the reports of previous works, as in the case of *Poerio et al. (2018)* where 51% reported watching videos daily or several times a week. This evidence suggests that a large proportion of the individuals who watched ASMR videos for the first time watch ASMR content again in a recurrent manner. Regarding the main motivations for watching ASMR videos, we found that relaxation was the most frequent reason for ASMR self-reporters followed by sleep induction (Fig. S5), which has been previously reported for people who experience ASMR (*Barratt & Davis, 2015*). Interestingly, in our study, the people who do not experience ASMR and have watched ASMR videos also reported relaxation as the main motivation for watching ASMR content. This result may suggest that ASMR content may have the potential to help people to deal with anxiety regardless of whether they experience the tingles or not. However, previous works have shown this positive effect may

occur only for individuals who experience ASMR (*Poerio et al., 2018*). Regarding the positive effect in sleep induction, ASMR triggers have been used to induce sleep in general population (*Lee et al., 2019*). The main findings of the present study constitute a starting point to continue investigating on the underlying mechanism mediating the association between ASMR and emotional regulation, which in the future could also open the possibility to use ASMR videos/triggers as tools to promote emotion regulation strategies in general population.

We also used a multiple regression with gender as the first predictor and showed that gender did not play an essential role in our main finding (see Supplemental Material). This result suggests that individuals who experience ASMR report higher scores on cognitive reappraisal regardless of their gender. Nonetheless, we had a limited capacity to assess the indirect effect of gender on the results due to the sample was composed mostly of women (in the ASMR and non-ASMR, women made 75.36% of each group). This unbalance is a limitation of the present study reducing our capacity to reach a more informed conclusion about the potential gender effects. Curiously, this more extensive representation of women seems common in other ASMR studies with self-reporting subjects. In a study by *McErlean & Banissy (2017)*, the percentage of women in the ASMR group was 69.88% and 80% in the non-ASMR group; in *McErlean & Banissy (2018)*, the percentage of women was 62.5% in the ASMR group and 72.47% in the non-ASMR group. Finally, in the study by *McErlean & Osborne-Ford (2020)*, the percentage of women was 74.19% in both groups. We believe that future studies on ASMR need to explore this tendency and also balance the proportion of female/male participants to precisely determine gender effects and potential gender differences in the experience of ASMR.

Finally, another limitation of the present study is related to the group selection's validity based on self-report over the internet. We grouped based on a YES/NO question widely used in previous studies (*Barratt & Davis, 2015*; *McErlean & Banissy, 2018*; *Poerio et al., 2018*; *Valtakari et al., 2019*). However, our description of the phenomenon was not very detailed (for instance, we did not provide examples of the triggers), potentially generating people confounding with other experiences, such as frisson. We then used complementary methods to corroborate that people who reported to experience ASMR actually did. We used the ASMR-15 scale, which has demonstrated effectiveness to measure the ASMR propensity in multiple dimensions of the phenomenon (*Roberts, Beath & Boag, 2019*), and also performed an additional analysis with only the people who sought ASMR videos. These results provided reliability to our group selection; however, this is still a big challenge to all ASMR studies performed over the internet. Some other methods have been used to ensure group selection is performed adequately (see ASMR checklist in *Fredborg, Clark & Smith (2017)*).

## CONCLUSIONS

This is the first study that examines whether ASMR self-reporters show differences associated with emotional regulation strategies compared with non-ASMR controls. We showed that people who experience ASMR use the cognitive reappraisal strategy to a

greater extent than non-ASMR people, suggesting more effectiveness in regulating emotions. The relevance of this finding relies on the fact that emotion regulation is fundamental for well-being, and this relationship between ASMR and emotional regulation may open the way to future research exploring the causal relationship between these features and also opening the possibility to use ASMR videos/triggers as tools to promote emotion regulation strategies, similar to how it is used to induce sleep (*Lee et al., 2019*).

Finally, our finding further elucidates individual differences related to this experience, supporting that ASMR is a real psychophysiological phenomenon associated with other psychological constructs and has remarkable consequences in affective/emotional dimensions and general well-being.

## ACKNOWLEDGEMENTS

We want to thank Pilar Fajardo for her significant help setting the online questionnaires and organizing data. We also want to thank Gissella Di Giovanni (Gisse ASMR) and Abigail (Abi ASMR) for their kind support and help in spreading the study's information.

### Funding

This research was supported by Fondecyt Postdoctorado No. 3180295 to Mario Villena-Gonzalez. The funders had no role in study design, data collection and analysis, decision to publish, or preparation of the manuscript.

### Grant Disclosures

The following grant information was disclosed by the authors:
Fondecyt Postdoctorado: 3180295.

### Competing Interests

The authors declare that they have no competing interests.

### Author Contributions

- Ricardo Morales performed the experiments, analyzed the data, prepared figures and/or tables, and approved the final draft.
- Daniela Ramírez-Benavides performed the experiments, prepared figures and/or tables, and approved the final draft.
- Mario Villena-Gonzalez conceived and designed the experiments, performed the experiments, prepared figures and/or tables, authored or reviewed drafts of the paper, and approved the final draft.

### Human Ethics

The following information was supplied relating to ethical approvals (i.e., approving body and any reference numbers):

The protocol was approved by the Ethics Committee of Pontificia Universidad Católica de Chile (approval reference number: 190325015).

## Data Availability

The raw data are available in the Supplemental File.

## Supplemental Information

Supplemental information for this article can be found online at http://dx.doi.org/10.7717/peerj.11474#supplemental-information.

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
