# Peer review of "Autonomous Sensory Meridian Response self-reporters showed higher scores for cognitive reappraisal as an emotion regulation strategy"

_PeerJ, doi:10.7717/peerj.11474_

## Round 0.1 · original submission · Major Revisions

· Academic Editor

Major Revisions

I have now received two reviews from experts in the field. Both reviewers are very positively inclined toward your manuscript, and I agree with their assessment. Both reviews are thoughtful, clear, and very detailed, so please refer to them while revising your manuscript. A successful revision needs some restructuring (e.g. some parts of the discussion moved to the introduction), the introduction of more details, and a better justification of the choice of the sample, the introduction of a small section on the possible limitations of the study.

Reviewer 1 ·

Basic reporting

1. Include citation for first sentence.
2. Missing full stop at end of last sentence of first paragraph of introduction.
3. Include citation for the creation of the term “ASMR”. I’ve included one that supports:

Morris, M. (2018). A simply marvellous reaction: Autonomous sensory meridian response and the desk tutorial. Architectural Design, 88, 50-57.

4. Line 69 – 71 – While I agree that there have been few attempts at establishing the prevalence of ASMR in the general population, there have been some recent studies that have offered estimates. I’ve included them below for your reference. It may strengthen your point to note the range of estimates when commenting on how widespread the phenomenon appears to be. In addition, it may be good to include a note/statistics to support your statement about the popularity of ASMR on the internet (i.e. google search trends etc.).

Poerio, G. (2016). Could insomnia be relieved with a YouTube video? The relaxation and calm of ASMR. In F. Callard, K. Staines, & J. Wilkes (Eds.), The restless compendium (pp. 119–128). Cham, Switzerland: Palgrave Macmillan. http://dx .doi.org/10.1007/978-3-319-45264-7_15

Roberts, N., Beath, A., & Boag, S. (2020). Autonomous sensory meridian response: sensitivity and personality correlates. Journal of Individual Differences. https://doi.org/10.1027/1614-0001/a000337

5. Line 74 - Consider including some citations to support the increasing interest in ASMR research.
6. Line 89 – Minor citation error re: et al.
7. Line 92 – Similar to point above, there has been limited research on the emotional correlates of ASMR propensity, however there have been a few studies that have measured the role of empathy in ASMR experiences. While this doesn’t directly address the relationship between ASMR and emotion regulation, I think it would strengthen your introduction to note the mixed findings.

Poerio, G. L., Blakey, E., Hostler, T. J., & Veltri, T. (2018). More than a feeling: Autonomous sensory meridian response (ASMR) is characterized by reliable changes in affect and physiology, PLoS ONE, 13 (6), e0196645.

(See supporting information).

8. Line 99 – I think the link between these two points could be improved with signposting.
9. Line 100 – Consider (as you’ve done in line 117) including some of the specific regions involved (in brackets).
10. Line 97-106 – I think the overall point of this paragraph (physiological and self-report evidence for emotional component of ASMR experiences) could be a bit clearer with some reorganisation.
11. Line 107 – “On the other hand” – I think your point here is actually consistent with the previous paragraph (i.e. ASMR being a complex emotional experience with arousal components).
12. Line 109 – Consider also including the following paper which assessed the relationships between self-reported misophonia and ASMR:

Roberts, N., Beath, A., & Boag, S. (2019). Autonomous sensory meridian response: Scale development and personality correlates. Psychology of Consciousness: Theory, Research, and Practice, 6(1), 22–39.

13. Line 119 – Minor - change the capitalisation of “It”.
14. Line 120 – I’m not quite sure what you mean by “such as the strategy deployed” – do you mean that ASMR may be a strategy/tool for emotion regulation?
15. Line 130 – Great that you noted the role of neuroticism, however, it would be good to also include it in the personality traits associated with ASMR in paragraph 4.
16. Line 147 – Consider rephrasing “ASMR self-reporters participants”.
17. Line 149 – Consider rephrasing “this is the first study aimed to”.
18. Line 156 – Consider opening the participants section with some more demographic information and details on where the participants were sourced.
19. Line 158 – Consider rephrasing “we randomly subsample”.
20. Line 165 – The ethics information may fit in better in the procedure section.
21. Line 168 – Missing full stop.
22. Just a suggestion, but it may be worthwhile moving some of the discussion of ASMR items (about videos etc.) to the materials section as a separate measure.
23. Line 194 – “10-item”.
28. Just a suggestion, but you may wish to move the first paragraph of the results (or just the reliability statistics) to the materials section (to compare prior and present reliability coefficients).
29. Consider including a subsection under “normality” to help streamline your results section.
30. Line 238 – Minor thing, just make sure you italicise the p and d statistics.
31. Line 238 – Make sure you report the t statistic to two decimal places.
32. Line 246 – “N” for total sample size, “n” for subsample.
41. Line 327 – Last sentence could be expanded, or omitted.
42. Line 338 – Consider including a citation here. I’ve included a reference for a study that assessed the effect of ASMR on sleep.

Lee, M., Song, C. B., Shin, G. H., & Lee, S. W. (2019). Possible effect of binaural beat combined with autonomous sensory meridian response for inducing sleep. Frontiers in Human Neuroscience, 13, 425. https://doi.org/10.3389/fnhum.2019.00425

Experimental design

24. Line 178 – Were all the participants Spanish speaking? Best to include this in the participants section, as well.
25. Participants - Were participants rewarded in any way for participating?
26. ERS - It would help the reader if you included the stem of the items, or some of the instructions that preceded the items, as well – i.e. how was the scale introduced?
27. It would be good to also include some reliability statistics of prior studies (i.e. Cronbach’s alpha) when you mention how the translated scale has been validated.
33. I understand that the ASMR qualitative responses were put into the supplementary materials, but I think it would be interesting to include a small discussion of the frequency that users engaged with ASMR media and the main reasons. Just a suggestion, but I think it may strengthen the story you are telling with regards to ASMR possibly being utilised as an emotion regulation strategy (as noted in the introduction).

Validity of the findings

34. Line 252 – What do these findings suggest? It may help the reader to briefly remind them what these two dimensions of emotion regulation refer to.
35. Paragraphs 2 and 3 – I think these paragraphs (aside from discussion of present findings) may fit better in the introduction to support the strong affective dimension of the experience, and the rationale behind exploring the relationship between ASMR and emotion regulation.
36. Paragraph 3 (Line 279) – This section may be better integrated into the first paragraph of the discussion, where you can first talk about the hypotheses and then what you found.
37. Paragraph 4 – I think your argument could be more clearly signposted here. Some further explanation would help to communicate how these findings relate to misophonia. It may also be worth noting the overlap in the incidence of ASMR and misophonia, where the conditions/experiences may occur in the same people. See the following paper for some discussion of the overlaps of these experiences, and some methodological issues encountered when attempting to disentangle them with respect to trigger context and individual differences:

Roberts, N., Beath, A., & Boag, S. (2019). Autonomous sensory meridian response: Scale development and personality correlates. Psychology of Consciousness: Theory, Research, and Practice, 6(1), 22–39.

In addition, the Rouw and Erfanian (2018) paper you cited earlier found a 49% overlap in the incidence of ASMR in misophonic participants. They may very well exist as extremes on the same spectrum of sensory-affective experiences, but they also appear to commonly co-occur. It would be good to see some brief discussion of this.

38. Line 303 – What does this suggest?
39. Paragraph 7 – While there are definitely imbalanced samples in ASMR research, the effect may be partially attributable to utilising university samples. It may also be worthwhile to note whether these group differences (by gender) were significant in previous studies.
40. Paragraph 8 – Good that you’ve noted some limitations, but it may also be relevant to note the potential limitations of the samples you drew from, confirmation bias, and limitations with self-report.

Additional comments

Thank you for inviting me to review your manuscript. The article expands on the current understanding of how ASMR relates to emotion regulation, and offers preliminary insight into the extent to which ASMR may reflect an emotion regulation strategy. The reviewer appreciates the breadth of related concepts the authors included in their discussion of ASMR experiences, particularly the clustering of both personality and physiological differences between ASMR-experiencers and controls. In summary, the paper appears to make an important contribution to our understanding of individual differences associated with ASMR propensity. However, I would advise that the authors refine sections of their manuscript to better emphasise this contribution.

I’ve made a number of suggestions and comments that I hope the authors will find helpful in refining the manuscript. Specifically, I would advise that the authors move some of the discussion material to the introduction. In particular, the prior research discussing the relationships between ASMR and other variables that weren’t included in the introduction. In addition, I appreciate the suggestion that ASMR may reflect an emotion regulation strategy, and I think it could be interwoven through the introduction a bit more explicitly.

Finally, while I understand that length may be an issue, it would be interesting to include some of the supplementary findings regarding motivation behind ASMR engagement to the results section. I think the results may provide support for the authors argument of the potential utility of ASMR as an emotion regulation tool.

·

Basic reporting

The writing in this manuscript is mostly clear and unambiguous (see relatively few comments in the accompanying annotated manuscript), with professional English used throughout. There are several typos, so the article requires a thorough copy-edit. The raw data was shared. A thorough overview of the background literature, complete with references, was also conducted, providing sufficient field background/context provided. I must commend the authors on an excellent job describing the current state of the literature on ASMR.

Although the article has a professional article structure, a formal discussion of the limitations of the study is lacking. Indeed, although the authors mention that their study cannot elucidate mechanisms and describe this as a limitation, they aren’t as forthcoming about the limitations of the study design (see section 2 on what I believe the limitations are).

Moreover, the authors describe frisson occurring in the context of musical experiences; although this is often considered the “primary way” folks experience frisson, this is not the only way. This should be amended.

Finally, the authors once again mention misophonia at the end of their paper, though don’t examine misophonia in their paper. I think it makes sense in the introduction, but this discussion should be more limited in the discussion section; see validity of findings section below.

Experimental design

Although the manuscript consists of original primary research within the aims and scope of the journal, there are a few key issues with the design of this study that must be addressed:
1. The authors only use one single question to determine the presence of ASMR. This is a major limitation of the study; the way the question is worded could be congruent with the experience of frisson/chills as well, and I imagine most lay folks won’t know the difference between ASMR and frisson. Indeed, I do not experience ASMR, but I experience frisson, and probably would have selected yes here. There are several measures of ASMR that have been used in the literature (e.g., the ASMR checklist by Fredborg et al., 2017, 2018) that could have been employed to ensure the ASMR-group truly experienced ASMR, and I wonder why the authors did not use such a measure. To put it another way, it is unclear whether the control/naïve participants are truly “naïve”. For instance, in an internet-based survey study of ASMR (Fredborg et al., 2017), potential control participants watched a 5-minute popular ASMR clip and completed a short survey to determine their eligibility in the control group. Although this method is not “perfect,” (i.e., perhaps some control participants truly experienced ASMR in other contexts), it supports the notion that this group is naïve to the ASMR experience.

I suggest that you run this analysis again with only the people who have ASMR and have actually sought out ASMR videos. That way, you have more evidence that these folks actually experience ASMR. You can add this as a follow-up analysis.

2. I also wonder if this was a biased sample, given that it appears that recruitment primarily took place over ASMR-related forums: “Participants were recruited by an online invitation to participate in the present work. This invitation was spread through social media, specifically through Instagram and Facebook, posted in different groups, either ASMR and non-ASMR related.” I am wondering if the authors had a biased sample toward ASMR based on their recruitment strategy. That said, I am happy to see that they used identical sized groups for the analyses. I’d like to see the authors discuss the limitations of self-report over the internet.

3. I am curious if the authors conducted any additional “validity checks” of the data. For instance, if questionnaires were checked over for folks who repeatedly answered the same response throughout, etc.

4. I understand that the authors used a random sample of the ASMR participants to compare to the control group. Did they try this multiple times (e.g., came up with multiple ‘random samples’ of ASMR participants and compared them to the control group)? If so, that would speak to the “robustness” of their findings and add more credence.

Validity of the findings

I believe this article mostly meets all criteria outlined by PeerJ with regards to validity. One issue is the following statement: “283 This finding provides further evidence supporting the notion that misophonia and ASMR 284 represent two ends of the same spectrum of sound sensitivity.” I believe the authors cannot make this claim based on the findings (the study only looked at ASMR vs. non-ASMR, not misophonics) and this should be revised.

Additional comments

Excellent work!

---

## Round 0.2 · Minor Revisions

· Academic Editor

Minor Revisions

Before accepting the paper I need you to address the final minor comments of one of the reviewers. The other reviewer is satisfied with your responses.

Reviewer 1 ·

Basic reporting

- Line 100: Minor error – Omit “Katherine” from citation
- Line 106: Minor suggestion – It might be useful for the reader if you include a couple of common misophonic triggers
- Line 122: Minor citation error with “J.”
- Discussion: Line 421: Missing brackets around “2018”
- Line 422-423: I think this sentence could be phrased a bit more clearly.
- Line 445-451: Minor, but be sure to use “and” for the out of brackets citations (as opposed to “&”)
- Line 459: Very minor, but instead of “proved”, it may be more accurate to say it has shown promise/demonstrated effectiveness, or something to that effect.

Experimental design

- Results: The discussion of the differences in ASMR groups by ASMR-15 scores was well done. However, was there an analysis of the total ASMR-15 score across groups, as well (i.e. sum of subscales)? If so, it would be great to include the results for the total score here, and the reliability of the total score measure in the materials section.

Validity of the findings

No additional comments here.

Additional comments

The reviewer commends the authors on their thorough and systematic integration of suggestions from the first review. Overall, aside from some minor suggestions, the reviewer believes that the paper has been substantially improved upon refinement. In particular, the emotion regulation focus has come through much more clearly in the introduction and discussion.

·

Basic reporting

Please see previous review - excellent work! There are still several minor typos that should be addressed (e.g., line 126 - "J" in citation, line 230 - google is in lowercase), but overall, very lovely work addressing mine and Reviewer 1's comments : )

Experimental design

Now that you've added in ASMR-15, this study is so, so much stronger and quite excellent. Wonderful work! All of my concerns were addressed.

Validity of the findings

I have no concerns here, all of my comments were addressed.

Additional comments

I love this paper. Thank you for contributing in such a meaningful way to the field! I'd love to perhaps do a replication here in Canada, if you are open to chatting! [email protected].

---

## Round 0.3 · accepted · Accept

· Academic Editor

Accept

I am pleased to inform you that your manuscript has been accepted for publication on PeerJ.